# Medical Relevance, State-of-the-Art and Perspectives of “Sweet Metacode” in Liquid Biopsy Approaches

**DOI:** 10.3390/diagnostics14070713

**Published:** 2024-03-28

**Authors:** Andrea Pinkeova, Natalia Kosutova, Eduard Jane, Lenka Lorencova, Aniko Bertokova, Tomas Bertok, Jan Tkac

**Affiliations:** 1Institute of Chemistry, Slovak Academy of Sciences, Dubravska cesta 9, 845 38 Bratislava, Slovakia; andrea.pinkeova@glycanostics.com (A.P.); natalia.kosutova@savba.sk (N.K.); eduard.jane@savba.sk (E.J.); lenka.lorencova@savba.sk (L.L.); 2Glycanostics, Ltd., Kudlakova 7, 841 08 Bratislava, Slovakia; aniko.bertokova@glycanostics.com

**Keywords:** cancer, glycan, sialic acid, diagnostics, liquid biopsy

## Abstract

This review briefly introduces readers to an area where glycomics meets modern oncodiagnostics with a focus on the analysis of sialic acid (Neu5Ac)-terminated structures. We present the biochemical perspective of aberrant sialylation during tumourigenesis and its significance, as well as an analytical perspective on the detection of these structures using different approaches for diagnostic and therapeutic purposes. We also provide a comparison to other established liquid biopsy approaches, and we mathematically define an early-stage cancer based on the overall prognosis and effect of these approaches on the patient’s quality of life. Finally, some barriers including regulations and quality of clinical validations data are discussed, and a perspective and major challenges in this area are summarised.

## 1. Current Challenges in Oncodiagnostics

Metastatic cancer remains, according to the WHO, one of the leading causes of global deaths alongside ischaemic heart disease and stroke [1]. Moreover, according to the Centre for Disease Control and Prevention, 70% of all medical decisions depend on laboratory tests results [2]; hence, the gold standard in oncodiagnostics continues to be histological evaluation of biopsied tissue. During the COVID-19 pandemic, we witnessed a decrease in routine access to clinical diagnostics due to quarantine restrictions, and it is very likely that another pandemic will occur in the future. Also, for many indications involving imaging techniques such as (multiparametric) magnetic resonance imaging (mp)MRI, long waiting periods and energy demands as well as CO_2_ production are significant factors [3]. Alternative approaches involving non-invasive, affordable and reliable cancer-specific tests are of high importance. Liquid biopsy tests offer promising solutions to the above-mentioned challenges, being able to provide information about the presence of clinically significant cancer due to the analysis of cancer-specific cell-free or circulating tumour DNA (cf/ctDNA), miRNA, small metabolites, proteins, extracellular vesicles and/or circulating tumour cells (CTC) [4]. Although often described as highly cancer-specific, the tissue specificity of these biomarkers might be questionable; thus, the same holds true for their use in a population-wide screening.

Post-translational modifications of proteins, mainly aberrant glycosylation [5], represent a promising biomarker, potentially integrating both of these specificities with a typical tissue-specific marker—PSA (prostate-specific antigen) in the case of prostate cancer (PCa) [6,7]. Ethical questions related to the use of laboratory-developed tests not yet involved in clinical guidelines should be considered. Any discomfort for the patient in the case of overdiagnosis/overtreatment (physical and mental, e.g., PSAdynia) [8] or false negativity/positivity also arising from biological variance in the presence/level of the biomarker should be investigated in detail. Primarily, the quality of life of the patient (QoL) is of the highest importance. The QoL parameter is strongly affected not just by certain conditions, but also by interventions with long-term consequences (as we propose in Figure 1). This pertains especially for asymptomatic indolent patients with PCa treated by radical prostatectomy, leading to erectile dysfunction and incontinence [9]. Overdiagnosis using ultrasensitive methods and markers might lead to unnecessary biopsies, such as in the case of PCa or breast cancers (BCa), creating a future barrier between patients and clinicians due to an unpleasant experience. Hence, setting a proper threshold for the analysis of biomarkers by every diagnostic method is a crucial step [10]. For innovative approaches, reproducibility and transparency in basic research, accessibility of retrospective clinical samples for some indications (such as early pancreatic cancer patient serum samples for pre-clinical validations) [11], recruitment for prospective oncological trials or (governmental) funding opportunities due to various commercial risks are also significant concerns in translational research [12,13,14].

### Future Decades and Human Health

In contrast with the above-mentioned PCa which occurs at a greater age and where, in many cases, only active surveillance is recommended, since the tumour is mostly localised and slow-growing [15], other oncological diseases require a prompt intervention. This pertains, for example, to small-cell lung carcinoma, triple-negative BCa, pancreatic ductal adenocarcinoma, glioblastoma and advanced ovarian cancer [16]. As in the case of PCa (e.g., urinal exosomal non-coding RNA) [17], for a more aggressive and rapidly spreading lung cancer, liquid biopsy tests are being developed and validated [18]. This holds great promise for their future use in clinical diagnostics—for the benefit of patients, but also for healthcare systems, as the costs per cancer patient might be almost doubled if cancer is diagnosed at a later stage [19,20].

In the current scientific literature, the glycoprofiling of proteins (i.e., partially or completely identifying a glycan structure covalently attached to a protein backbone) is not usually listed among “liquid biopsy” approaches, despite the fact that such assays can indicate that attention needs to be paid to a specific organ or a group of organs and identify the process of tumourigenesis at an early stage. Another important aspect of liquid biopsy is to provide information about the stage of the disease and a prognosis for the patient’s survival, possibly a design for a personalised therapy plan [21]. Herein we present the current state-of-the-art for glycoprofiling of novel glycoproteomic biomarkers, mainly focusing on sialic acids (*N*-acetylneuraminic acid—Neu5Ac—and *N*-glycolylneuraminic acid—Neu5Gc—respectively) in early cancer diagnostics as a possible multi-omics liquid biopsy approach for the analysis of prevalence and also for more aggressive cancer types, such as non-small-cell lung cancer [22,23]. The importance of these methods is further illustrated by the fact that the global incidence and mortality for different cancer diseases is predicted to grow in the upcoming years. The predicted growth (in %) from 2020 to 2040 for the commonest solid tumours is shown in Figure 2 [24].

Liquid biopsy, offering “a universe in a vial of blood” as described in ref. [25], uses only a minute amount of sample, and the greatest risk it possesses is a possibility of false positive and false negative results, leading to stress for the patient, further testing and a possibility for further development of cancer, respectively. These risks, however, are associated with every single laboratory method; hence, only by combining analysis of different biomarkers, methods and approaches, is sufficiently high diagnostic accuracy achieved for resectable tumours [26,27]. Compared to imaging methods currently in use, the result of a laboratory test is mostly quantitative, measured objectively and evaluated as a part of a larger pool of analytes available. Medical images, being very complex, are interpreted by humans, and thus are subjective [28]. In this review, we aim to critically evaluate the possibilities of using sialic acid-terminated glycans (complex, often branched saccharides) in modern liquid biopsy approaches as well as the analytical platforms suitable for routine screening in clinical practice.

## 2. “Unpredictable” Metacode of Life—Biochemistry of Sialoglycoconjugates

Glycosylation is the commonest co- and post-translational modification of proteins in eukaryotic cells [29,30,31,32]. At the same time, the majority of extracellular and serum proteins are glycosylated, as well as the most commonly used oncomarkers, such as CEA (carcinoembryonic antigen), AFP (α-fetoprotein) or PSA (prostate-specific antigen). Through en bloc attachment, trimming/extension and terminal glycosylation in Golgi, glycosylation is driven by glycan-modifying enzymes (GME), namely hydrolases and transferases [33]. The final glycan structure is, however, not possible to predict directly from a genetic code, but rather is dependent on several factors, such as (i) expression of GME, (ii) availability of precursors (activated monosaccharides) and (iii) Golgi fragmentation [34,35]. Availability of the activated precursors (nucleotide sugars) is closely connected to glucose metabolism, which in turn, is known to be significantly shifted in cancer due to the Warburg effect (observed by Otto Warburg et al. in about 1920 and published later) [36]. The Warburg effect, a shift toward anaerobic metabolism and increased lactate production in the presence of oxygen and functional mitochondria, is inefficient in generating ATP as an energy-source-providing molecule; however, it may store unrespirated carbon inside the cells for further growth [37]. Furthermore, lactate promotes many essential processes, such as invasion, immune escape (acidification of extracellular pH to 6.3–6.9, leading to apoptosis of natural killer and natural killer T cells), metastasis and angiogenesis [38,39]. Glucose uptake (by glucose transporters) is strongly increased in cancer cells, leading to possible starvation in neighbouring cells [40], while changes in mTOR, p53 or KRAS signalling lead to an accumulation of glycolysis intermediate products [41,42], shifting metabolism towards the pentose phosphate pathway (generating precursors for nucleic acid synthesis and mitigation of ROS, reactive oxygen species, effects) [43] and hexosamine synthesis [44,45]. The latter might affect the glycan composition and shape, in combination with the expression and availability/activity of GMEs. In addition, pyruvate kinase PKM2 (muscle isoform) catalysing a rate-limiting step in glycolysis (conversion of phosphoenol to pyruvate) has been shown to be translocated into the nucleus, where it serves as a kinase for other protein targets, promoting several pathological processes [46,47]. A schematic presentation of these mechanisms and their effect on aberrant glycosylation is shown in Figure 3.

### Three Levels of Regulation—Genes, Substrates and Golgi

Unlike proteins, the glycan structure cannot be directly predicted from any template, such as the genetic code, but depends on many factors directly connected to genetic factors such as GME expression, hence the denotation “metacode”. Often described as the third alphabet of molecular biology, it remains unclear whether these carriers of biological codes yield information which is universal in determining an organism’s fitness [48]. An abundant component of these glycans in animals, decorating their terminal structures, is sialic acid. Although sialic acids are actually a group of nine-carbon saccharides derived from neuraminic acid, these days, the term is most commonly used as a synonym for Neu5Ac (2-keto-5-acetamido-3,5-dideoxy-D-glycero-D-galactononulopyranos-1-onic acid) found in humans. As posited by Darwin or Huxley, great apes are our closest evolutionary cousins, but there are some minor differences at the genetic level [49]. One of these changes comprises an inactivating mutation in the CMP-*N*-acetylneuraminic acid hydroxylase (*CMAH*) gene, resulting in accumulation of the precursor of Neu5Gc (NGNA) synthesis in humans. Neu5Gc present in humans is thus closely linked to increased dietary uptake [50,51], for instance, present in red meat [52] and possibly other food matrices. Food glycomics research, especially the interaction of the human gut microbiome with glycome contained in food matrices, such as milk, is currently on the rise [53]. Sialic acids in vertebrates are abundant on cell surfaces, being an integral part of membrane-bound mucins or gangliosides (some of which are even disialylated, such as GD2 (tumour-associated glycan epitope), recently identified as a BCa stem cell marker that promotes tumorigenesis) [54], contributing to cell–cell communication and/or adhesion processes. Sialic acids bound to proteins in different ways also form a common part of soluble glycoproteins released into the environment. Neu5Gc, despite its negative effects on human health such as correlation with chronic inflammation-mediated autoimmune diseases or cancer (e.g., BCa or ovarian cancers) [55,56,57], continues to be detected on various approved and marketed biotherapeutics or xenografts [58]. The glycoengineering of therapeutic glycoproteins is, accordingly, a critical part of novel drug development and needs to be considered prior to any future approval of cytokines, enzymes or immunoglobulins for therapeutic purposes [59]. Furthermore, sialic acid-binding lectins and modifying enzymes, sialoside inhibitors and anti-Siglecs (sialic acid-binding immunoglobulin-like lectins) antibodies are known to be druggable targets for immunomodulatory purposes [60]. These facts highlight the relevance of sialic acids in human health and of monitoring the changes in sialic acid content and structure in different physiological (ageing) as well as pathological (inflammation, infection, cancer) processes for a better understanding and potential use in future clinical diagnostics.

**Figure 3 diagnostics-14-00713-f003:**
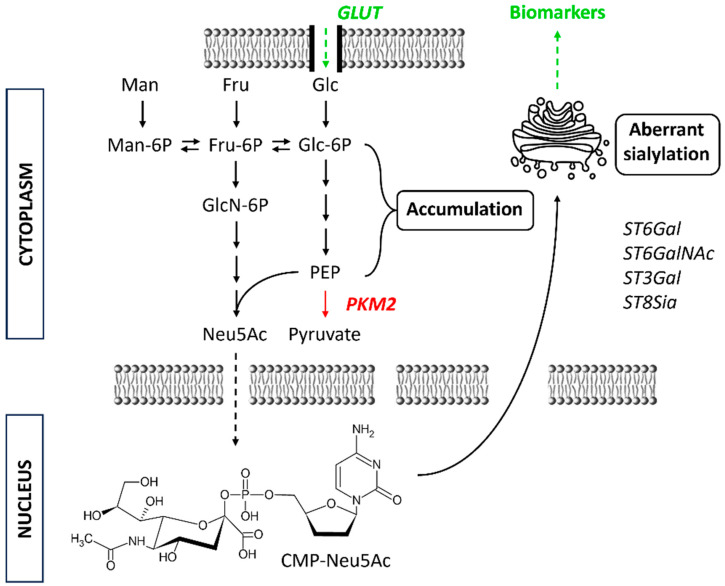
Schematic presentation of glycolytic pathway alteration during tumourigenesis as a part of Warburg effect. This leads to increased CMP-Neu5Ac levels (precursor) and together with certain sialyltransferases (*ST*s) results in aberrant sialylation of glycoproteins. Fru = fructose; Glc = glucose; GLUT = glucose transporter; Man = mannose; Neu5Ac = N-acetylneuraminic acid (NANA); PKM2 = pyruvate kinase muscle isoenzyme 2.

In eukaryotic cells, Neu5Ac is synthesised in the cytosol from hexosamine pathway precursors, transferred to the nucleus for activation by enzyme CMP-Neu5Ac synthetase and, in the form of CMP-Neu5Ac, subsequently transferred to the Golgi [61]. Here, sialyltransferases alongside other glycosyltransferases form differently glycosylated glycoconjugates. The expression of these sialyltransferases might be dysregulated as a result of tumorigenesis [62]. Overexpressed sialoglycans, such as sialylated Lewis antigens (sLe^X^ and sLe^A^), are known to promote tumour metastasis. Disialyl-T glycan, possibly regulated on the surface of cancer cells by dysregulated oncogene *MYC*, leads to immune evasion [63,64]. These glyco-immune checkpoints suppressing anti-tumour reactivity by interacting with immunoregulatory Siglec receptors on myeloid and lymphoid immune cells are of great importance in cases of glycosylated CD24 or CD47, being a phagocytic inhibitor, e.g., for cancer stem cells [65,66]. Figure 4 shows schematic changes in *N*- and *O*-glycosylation accompanying tumour formation and progression. Sialylation and hypersialylation often lead to immune system processes’ modulation, as in the case of factor H recognising sialoconjugates and modulating complement activation [67], or the presence of α2,8-Sia containing structures protecting a tumour from the cytotoxic effects of NK (natural killer) cells [68]. Truncated sialylated *O*-glycans, on the other hand, were proposed as a target for novel anti-cancer immunotherapy; however, the main obstacle in this area is a low specificity and cross-reactivity [69]. There is a fourth, much less common and known sialic acid linkage—α2,9-linked sialic acid—detailed by Miyata et al. in 2006, found in nature as a sulphated polysialic chain isolated from sea urchin sperm flagella glycoprotein denoted as flagellasialin [70].

Except for sialic acids, heparan sulphates make up the outermost part of cellular surfaces, carrying a negative charge involved in a membrane fixation of various ligands, including viral particles [71]. For SARS-CoV-2 it was shown, however, that neuraminidase treatment increased the ACE2–spike protein interaction; hence, sialic acids might play an important role in the biorecognition of SARS-CoV-2 similar to the sialic acid-triggered recognition of the influenza virus [72]. Sulphated glycans attached to a protein backbone, such as syndecans or glypicans; proteoglycans, members of the so-called heparan sulphate proteoglycans family, are well-known examples of negative charge carriers with an impact on viral adsorption and penetration, but also for cancer cell proliferation and growth [73,74], even inducing a pro-migratory effect in BCa cell lines due to the presence of a positive charge in the ligand structures [75]. Other examples of sialylated glycans assisting the inactivation of CD4+ T cells and macrophages, inducing a tolerogenic effect in immune cells and potentiating cancer metastases, are tri- and tetra-antennary *N*-glycans, Tn and sTn antigens present in *O*-glycans, and poly-*N*-acetyllactosamine chains present in *N*- and *O*-glycans, respectively [76].

## 3. Techniques for Mapping the Human Oncosialome

Analysis of disialyl-T and glycans with similar functions, i.e., which play an important role in immune evasion, can be used in “early diagnostics”—a precondition for any liquid biopsy approach. Tumour growth is a biphasic process consisting of initial exponential growth followed by linear growth. As shown in Figure 5, when a cell undergoes a transformation and becomes a cancer cell (C), it starts to compete for resources with the surrounding tissue. This so-called logistic model produces an S-shaped growth curve [77]. The growth rate (*r*) and death rate (*μ*) are different for both cases. A tumour (clinically significant, cs—meaning a tumour which has surpassed the exponential growth phase in the sigmoid growth curve and is now growing linearly) develops once *r*_2_ > *r*_1_ (the equilibrium is disrupted). This is due to several factors, one of which being the expression of cancer-related glycans, G(C), inhibiting the reaction of immune cells (IC). The diagnostics aims to detect the state when the cancer cells persist, i.e., *μ*_2_ ≤ *r*_2_. Cancer cells and the surrounding tissue (T) still co-exist in the same medium, hence the term (T + C)/K, where K is a constant—the carrying capacity of the environment, determined by the available resources. To overcome this drawback and secure nutrients and oxygen supply, the tumour produces angiogenic factors, such as a vascular endothelial growth factor, which not only induces vasculatures’ formation, but also creates a basis for future metastases’ dissemination [78]. Aberrant glycosylation has been shown to promote tumour angiogenesis by extracellular matrix degradation. This activates the angiogenic signalling pathways [79]. Without vascularisation, the cells exposed to hypoxic stress activate hypoxia-inducible factor 1α gene (*HIF-1α*), significantly affecting glycosylation—including sialylation through the epimerisation of UDP-GlcNAc (GlcNAc = *N*-acetylglucosamine) by UDP-GlcNAc 2-epimerase. This process contributes to oncogenic transcription factor stabilisation and drives cell-surface biomolecule (hyper)sialylation [80].

Although more complex, if aberrant glycosylation forms a part of an early clinically significant tumour development, detecting these aberrant signals might be a sign of early cancer development and a prerequisite for in-time and, thus, less invasive interventions, such as basic surgery. In this chapter, methods for early and non-invasive (i.e., liquid biopsy-based approaches with no need for tissue biopsy) techniques developed and/or proposed are summarised. Although single molecule glyco-analysis using low-temperature scanning tunnelling microscopy has been performed only very recently, more user-friendly and clinically compatible methods are of great importance for a routine evaluation and detection of aberrant and cancer-related glycans [81]. The amount of information needed is not necessarily too complex, but the presence/absence of some specific glycan epitopes might be sufficient in the case of a multi-omics approach; thus, lectin-based technologies might be of great importance in common immunochemistry/ELISA formats.

**Figure 5 diagnostics-14-00713-f005:**
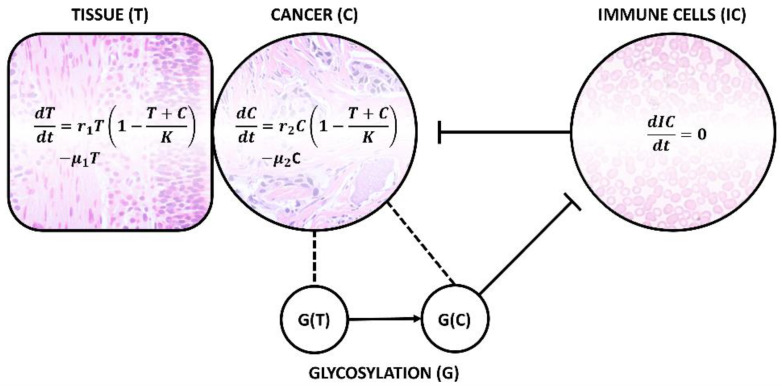
Schematic presentation of logistic mathematical model for cancer cell (C) growth in the same medium as healthy tissue (T) and negatively interacting immune cells (ICs, with more or less constant concentration) unless aberrant and cancer-specific glycosylation takes place, in which case, novel G(C) glycan expressed on the cell surface causes a phenomenon known as immune evasion. As a consequence, death rate *μ* becomes lower than the growth rate *r* and cancer cells persist, producing angiogenic factors and securing nutrients and oxygen in competition with its surroundings. Hence, these processes are an early sign of tumour development.

In general, the dynamics of cancer evolution could be described by different mathematical models—most commonly by ordinary differential equations (ODEs), but also by partial differential equations, algebraic equations or empirical models. Many of these models attempt to account for tumour heterogeneity, such as by the presence of proliferative and quiescent cells, sensitive and resistant cells or by heterogeneity on the gene level. Genes, however, only represent the first step towards unveiling a greater picture of the biomolecular diversity of tumours at both proteomic and glycomic levels [81]. The most commonly used models include linear growth (constant zero-order growth), exponential growth (first-order growth proportional to tumour burden), and the expansion of these models by first-order shrinkage, taking into account natural tumour cells’ death rate and, finally, logistic and Gompertz growth—both taking into account the realistic scenario of decreasing the growth rate over time as the tumour burden increases [77]. The current knowledge of tumour cell growth rate and proliferation kinetics depends on the measurement of the cell’s doubling time. These rates vary greatly depending on the specific tumour and its cell type and are correlated with the histological type. In general, the doubling time-affecting factors are (i) cell cycle duration, (ii) proportion of proliferating cells and their birth rate and (iii) cell loss factors [82]. Tumour growth kinetics is also an important prognostic and predictive marker, being evaluated in cases of a locally advanced non-small-cell lung carcinoma as inversely proportional to progression-free survival, i.e., higher tumour growth rate prior to any treatment refers to a poor prognosis [83]. Early-stage diagnostics and in-time intervention thus increase the survival rate by removing the tumour while still localised to one organ. Hugh J. Freeman defines early-stage (colon) cancer as “disease that appears to have been completely resected with no subsequent evidence of involvement of adjacent organs, lymph nodes or distant sites”, i.e., with no metastases yet present [84]. Once the cell activates its immune evasion mechanisms (such as sialoconjugates expressed on its surface), affecting its own survival rate and due to angiogenesis and its growth/death rate at the expense of the surrounding tissue, the tumour starts to show signs of clinical significance and increases its metastatic potential. Early detection significantly influences the expenses related to cancer cure in oncology. In cases where a population consists of already diagnosed patients with early-stage cancer, diagnostic/imaging methods and surgery are supported. In cases of metastatic disease being present, investments in palliative care are necessary [85].

### 3.1. Complexity of Glycans as a Hope and Barrier in Early Diagnostics

Sialo-oncomarkers have been known for many years but, due to huge complexity in the structures present in biological samples and their biological variability, their role in cancer development and/or progression is not fully understood. Figure 6 depicts the commonest sialic acid structures, namely G_D2_, G_D3_ and G_M2_, along with sTn antigen and sialyl-Lewis (sLe) antigens. sLe^A^ (CA19-9) is an FDA-approved oncomarker, a tetrasaccharide usually linked to *O*-glycans containing α2,3-linked sialic acid, commonly recommended for the management of patients with pancreatic cancer (pancreatic ductal adenocarcinomas), but not suitable for screening purposes [86]. The same holds true for CA125, a mucin-16 molecule used for the management of metastatic ovary carcinomas with rather low AUC value (from the ROC—receiver operating characteristic curve) of 0.632 [87]. This complexity of glycans enhanced by different types of glycosidic bonds and possible branching makes them exceptionally difficult to analyse in a routine manner. Highly skilled operators and precision are needed for the entire process, including data interpretation. Two main approaches may be used for the analysis, i.e., (i) “heavy machinery” such as mass spectrometry, liquid chromatography, capillary electrophoresis and/or their combination and (ii) the use of glycan-recognising molecules, such as antibodies, lectins—glycan-binding proteins of non-immune origin, aptamers, synthetic receptors, engineered neolectins and boronic acid derivatives [88,89,90,91]. This latter approach enables the development of more clinically compatible detection platforms, such as ELLBA (Enzyme-Linked Lectin Binding Assay) [92] or microarray format [93,94]. The method can be further divided as (i) label/label-free, (ii) with/without glycan release (in situ glycoprofiling) and (iii) with/without any chemical or biochemical treatment (i.e., derivatisation and enzyme treatment).

Commonly used methods in the analysis of glycans involve mass spectrometry (MS), often in combination with various other analytical tools, such as chromatography. As MS is unable to determine the type of glycosidic bond based on the mass-to-charge ratio analysis of the ionised fragments alone, chemical or biochemical methods in the pre-analytical stage (i.e., derivatisation or enzymatic modification) are necessary, as well as permethylation of the isolated glycans increasing ionisation efficiency and stability of the analysed glycans, which possess a high level of microheterogeneity [95]. Sialidases (enzymes able to selectively distinguish and hydrolyse a specific type of glycosidic bond between sialic acid and galactose residues) are used to distinguish between α2,3- and α2,6-bonds when used in combination. Sialylated glycans were recently shown to be enriched in a Ti_3_C_2_T_x_ MXene-based (2D inorganic nanomaterial, generally carbides, nitrides or carbonitrides of various transition metals) cartridge prior to MS analysis [96]. Apart from these and other exoglycosidases, PNGase F (cleaving the bond between the innermost GlcNAc and asparagine residues) and endoglycosidase H (endo H, cleaving the bond between the GlcNAc residues of the chitobiose core in high mannose and hybrid glycans) can be used to release the glycans from proteins for further enrichment strategies, such as hydrophilic interaction liquid chromatography (HILIC) and electrostatic repulsion liquid chromatography (ERLIC) [97,98]. Also, different enrichment strategies prior to analysis, such as the use of lectins with different specificities or TiO_2_ for sialic acid-containing glycopeptides or glycoproteins, are often used [99]. Linkage-specific chemical derivatisation involves methods distinguishing the type of glycosidic bonds based on the products with different masses, such as lactonisation/dimethylamidation, of the analysed fragments [100,101]. Other approaches involve methylester, ethylester, amide or isopropylamide formation in the case of α2,6-bound sialic acid [102]. The derivatisation of glycans can result in the loss of sialic acid residues during sample pretreatment, permethylation, and labelling of the reducing end of glycans. Accordingly, glycan processing needs to be performed carefully to yield as much information as possible about the native structure of the respective glycan. Mass spectra interpretation might be further complicated by the fact that sialic acids form salts (-COOH group forms -COONa, -COOK, etc.) with different masses, leading to the presence of multiple peaks for one epitope [102].

**Figure 6 diagnostics-14-00713-f006:**
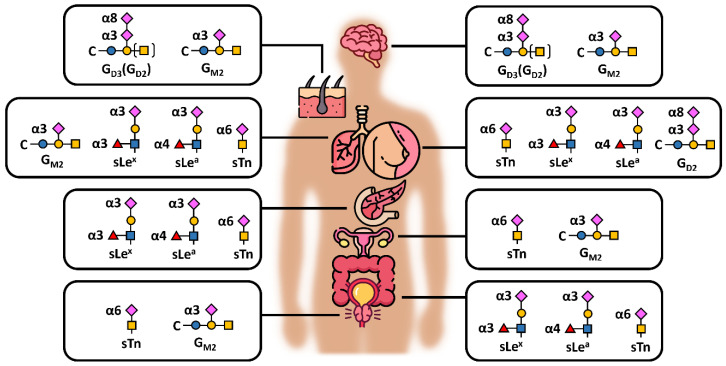
Schematic presentation of different sialic acid-containing epitopes/antigens in the commonest human carcinomas (excluding haematological malignancies, i.e., from top to bottom: brain, skin, lung, breast, pancreas, ovaries, colon and prostate), showing also different glycosidic bonds between sialic acid and underlying monosaccharide—namely α3, α6 and α8; s = sialyl, Le = Lewis antigen, C = ceramide. The structures were drawn based on data available in several publications [69,103,104,105,106].

### 3.2. Alternatives to Mass Spectrometry in Glycan Analysis

Lectins, on the other hand, can often be used in a miniaturised format of analysis with a minute sample consumption and high throughput, such as microarray [94,107], lectin-based ELISA (ELLBA) [108] or as a part of biosensors [109,110] with different transducers, i.e., electrochemical, optical, etc. Two main drawbacks of lectins as analytical tools, namely (i) a lack of analytical sensitivity due to low binding affinity (high K_D_ values) and (ii) limitations in availability of lectins recognising less common, disease-specific structures or rare epitopes, such as Neu5Gc, were formulated by Haab in 2012 [111]. Two additional problems involve the so-called “promiscuity” of these molecules and lectin specifications published by different suppliers, and finally their use in a sandwich format of analysis. Lectins often recognise a variety of different glycans which are structurally similar, their affinity being affected not only by the terminal monosaccharide unit but also by the second or even third monosaccharide in the chain, such as in the case of *Maackia amurensis* agglutinin and its two isoforms [112]. In some cases, lectins and the information they yield can easily be misused or misinterpreted if they bind to structurally distinct epitopes, such as in the case of WGA (wheat germ agglutinin), which binds to GlcNAc but also to Neu5Ac-galactose epitopes [113,114].

When using lectins (and especially sialic acid-binding lectins) in the analytical technologies listed above, for an easy and user-friendly manipulation of the suggested diagnostic kits, direct use of a biological sample (with minimal to no sample treatments) should be a priority [115]. For the purpose of liquid biopsies, the tissue specificity of the proposed biomarker should always be ensured by choosing an appropriate antibody enriching only a single protein from a sample for further glycoprofiling. Enrichment of the glycoprotein of interest on the surface of the plate well or magnetic particle is followed by glycoprofiling using a lectin conjugated to a label (fluorophore or HRP enzyme—horseradish peroxidase). There are, however, methods using no labels at all, such as optical and electrochemical biosensors [99,116]. The two main issues with this sandwich configuration involve (i) a possible cross-reaction of the lectin with the underlying antibody (immunoglobulin gamma, IgG) containing in its Fc fragment two biantennary *N*-glycans at Asn297 (see Figure 7), commonly terminated with sialic acid and (ii) blocking the surface against non-specific interactions, as the blocking agent might contain impurities promoting these interactions. As there are many commonly used blocking agents varying substantially in their composition, additives content or purity, the cross-reaction of lectins (and even some other proteins) with some of these might be an issue [117]. This cross-reactivity issue can be mitigated or eliminated by a mild NaIO_4_ oxidation protocol introduced in 2007 for IgG glycans, followed by the attachment of hydrazide derivatives to the aldehyde groups formed [118]. This, however, leads to an increase in K_D_ values for the Ab-Ag binding pairs due also to the affection of amino acid side chains [27]. Gentler conditions can be used in cases where only sialic acid needs to be oxidised compared to other structures buried deeper inside the glycan chain, i.e., the use of 25–150 mM NaIO_4_ for glycoprofiling using SNA and Con A lectins, respectively.

### 3.3. Glycosylation of Immunoglobulins in Diagnostics and Therapy

Despite the fact that the glycosylation of IgG is inherently a limiting factor when used together with some lectins, IgG glycans play an important role in the antibody’s nature—whether the antibody acts as a pro-inflammatory or anti-inflammatory agent. The trimming of sialic acid and galactose units from the *N*-glycan’s non-reducing end leads to greater binding of the mannose-binding protein to these structures, activating a complement via a so-called lectin pathway [119,120]. Mannose, being more accessible for biorecognition due to the loss of one or two galactose residues in the biantennary structure (occurrence of so-called G0 and G1 IgGs), which also lowers IgG’s half-life in the bloodstream, is positively correlated with the onset of rheumatoid arthritis and osteoarthritis, as previously shown in a pioneering paper by Parekh et al. in 1985 [121]. Furthermore, monoclonal IgG produced in non-human systems might contain glycan structures different from those normally found in the human body [122], affecting the therapeutic effect of a drug administered to a patient, causing immunogenic issues. Novel “glycoengineered” antibodies (amongst other proteins, such as cytokines and enzymes) include non-fucosylated obinutuzumab (Gazyva^®^) binding more efficiently to the Fcγ receptor IIIa present on immune effector cells than to non-fucosylated mogamulizumab (Poteligeo^®^) [59,123,124].

Although there are some cases where the functional significance of a glycan structure for human health has been revealed, there remains a need to decipher the “glycocode” (or sugar code, as has been repeatedly proposed by H. Gabius since 1998) [125,126,127] for detecting several pathological processes. Figure 8 shows various analytical configurations for in situ detection of glycans (i.e., without the glycan being released from a protein backbone prior to the analysis) using lectins or glycan-binding proteins, such as antibodies which, unfortunately, are still in limited supply [128]. Without using an antibody monolayer during the glycoprofiling process, no tissue specificity can be guaranteed. Although some studies suggest that a whole-serum level glycoprofiling might be used for diagnostics of some cancers, interference with other comorbidities and thus higher false positive results might be an issue. Complementary methods based on this approach have been published for primary and metastatic brain tumours [129], hepatocellular [130], breast [131], colorectal [132] or lung cancers [133]. In addition, comparing healthy serum glycome from individuals of different ethnicity and/or age yields important information about the baseline of physiological structures present in these individuals, helping to search for pathological processes occurring subsequently, making this approach a useful tool in the area of biomarker discovery [134,135]. Some of the configurations take advantage of the use of magnetic particles offering high-throughput, low costs, energy consumption efficiency, increased disease specificity of the assay and sensitivity [136]. An important aspect to be taken into account in designing the diagnostic assay is the peroxidase-like activity of magnetic nanoparticles possibly affecting the background signal in colorimetric assays [137].

**Figure 7 diagnostics-14-00713-f007:**
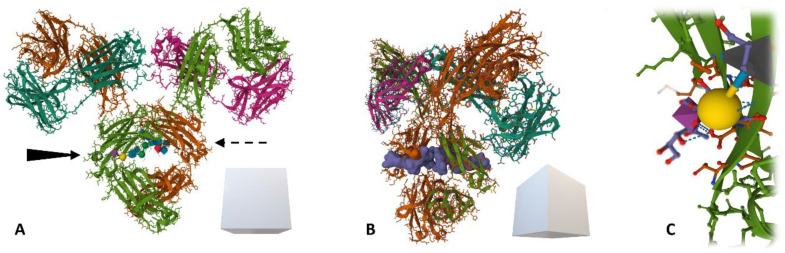
A ribbon model (with amino acid side chains) of human immunoglobulin (IgG) (source: protein data bank, pdb code: 1MCO) with two conserved *N*-glycans at Fc fragment’s Asn297. Brown and grass-green = two heavy chains, dark green and magenta = two light chains. (**A**) IgG structure—frontal view, with depicted sialic acids on both sides, (**B**) semi-profile of the IgG with a Gaussian surface molecular model of the glycan (blue) accessible for biorecognition and (**C**) detail of sialic acid bound to galactose residue via α6-glycosidic bond recognised for instance by SNA-I isolectin.

**Figure 8 diagnostics-14-00713-f008:**
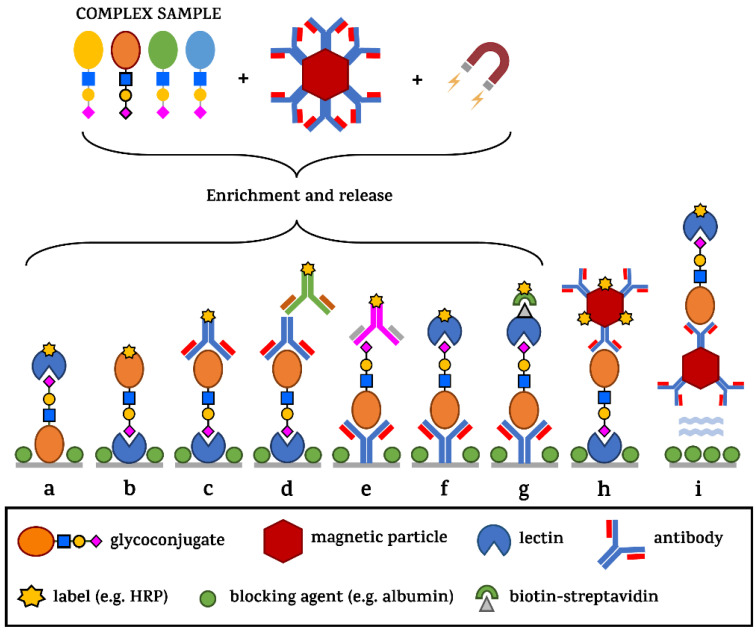
Different sensing configurations commonly used in ELLBA and/or microarray formats. From a complex sample, the analyte of interest can be extracted from the sample by Ab-modified magnetic particles, and then released (a–g) or used in a direct glycoprofiling format (h) on a well-bottom or entirely in solution—without any molecules being immobilised on the ELISA plate surface—except for the blocking agent inhibiting non-specific surface binding (i). The signal generation in the case of ELLBA takes advantage of the amplification by enzymes linked to a biomolecule or to streptavidin used in combination with biotinylated recognition elements.

## 4. Perspectives of Glycan Liquid Biopsies and Clinical Validations

Glycans in the form of different glycoconjugates as a novel type of diagnostic biomarker are more readily obtainable than other biomarkers, such as miRNAs, ct/cfDNA/RNA, extracellular vesicles including exosomes and circulating tumour cells, as they might be more abundant in body fluids and do not require any amplification, unlike nucleic acids (which are all comparably simple to obtain) [138]. While miRNAs usually need Northern blotting, in situ hybridisation methods using locked probes, microarray approaches, next-gen. sequencing or real-time quantitative PCR [139], the detection of glycans can be performed within a short time period using only whole serum with no special pre-treatment and an ELLBA approach. As cancer diagnostics medical devices are regulated by the FDA as class III devices due to the possible harms caused by false positive or false negative results, these methods need to be highly reliable, reproducible and robust, with sufficient clinical evidence with regard to their performance [140]. This is most probably why miniaturised, easy-to-fabricate, portable and cheap biosensors are not routinely used for cancer screening. Other obstacles to liquid biopsy cancer screening aiming to detect any pre-malignant lesions for an early intervention include geography, services coordination, accessibility, medical guidelines and recommendations and health insurance [141]. Challenges in liquid biopsy mainly involve technical complications in the isolation process of CTCs, lack of knowledge about tumour-specific variants of cfDNA, low ctDNA concentrations in the early stages of a disease or accurate quantification of scarce and small-size miRNA in body fluids [142]. Age, gender, ethnicity and disease history might also affect the quantity of circulating miRNA [143]. The situation is even more complicated in the case of exosomal miRNA, where exosomes have to be isolated from a pool of other extracellular vesicles, such as microvesicles with a size similar to that of exosomes in their lower size range [144,145]. This issue can, however, be readily overcome by a proper isolation technique, such as affinity-based methods involving highly specific antibodies against exosome-specific surface receptors (such as tetraspanins CD63, CD81, etc.), leading to a population of exosomes more uniform than that achieved by ultracentrifugation [146]. A comparison of miRNA and selected sialoglycans to detect a specific solid tumour based on an AUC parameter (obtained from ROC analysis, see section below) is summarised in Table 1.

### Performance of Glycans in Clinical Practice

In verification and validation studies detailing new clinical biomarkers, a receiver operating characteristics curve (ROC) is commonly used, with the area under the curve (AUC) parameter as the main indicator of the biomarker/assay performance. Sensitivity (true positive rate) and specificity (true negative rate) are extracted from the ROC based on a desired or optimal threshold and overall diagnostic accuracy, and negative and positive predictive values are calculated. The threshold for oncodiagnostics should always be set at a sufficiently high sensitivity level (e.g., 90–95% sensitivity), even at the cost of low specificity possibly leading to unnecessary biopsies (as in the case of PCa diagnostics), as missing the tumour during the examination might lead to disease progression and ultimately the patient’s death. According to Verbakel et al., a threshold based on a desired combination of specificity and sensitivity also has a direct relationship with the relative costs of false positives and false negatives; if an individual should be treated when the risk of an event is 10% (1:9), correctly treating one patient with an event (disease) justifies an unnecessary treatment for nine healthy individuals [147]. Furthermore, when using ROC for data evaluation, all of this method’s limitations must be considered, such as: (i) ROC does not account for prevalence (proportion of people with a disease at one particular point in time; not the same as incidence); (ii) ROC does not account for misclassification of patients due to, for instance, false negative biopsies or falsely elevated serum levels of some biomarkers; (iii) ROC requires the data to be properly balanced (by way of their count/quantity and quality as well) to mitigate some significant effects such as age, gender, ethnicity, comorbidities, etc. (which may lead to over-optimistic performance); (iv) ROC requires a normal distribution of input data and cohort sizes comparable with the population studied. When combining several marker types which have already been shown to increase the overall clinical significance [26], the lowest possible number of these markers should be maintained to preclude overfitting and cost-effectivity of diagnostics. Although the ROC curve is most suitable in the early stages of clinical research and verification of a proposed biomarker, a change in the AUC parameter has little to no meaning for clinical practice. Other methods highlighting the net benefits for patients should be introduced at later stages, as the ROC curve does not allow for clinical interpretation of the data [148,149].

Although glycomics has already reached clinical practice in the case of glycoengineered therapeutic proteins or even a diagnostic test (Glyco Liver Profile, Helena Biosciences) [150], some obstacles have to be overcome if wide use of these methods is to be expected in the near future. These barriers can be summarised as follows: (i) a lack of glycoconjugate analytical standards or calibrators in the case of diagnostic tests; (ii) limited access to so-called high-throughput glycomics in clinical practice, while at the same time these methods are largely focused solely on *N*-glycans and rely on a skilled operator and databases based on long-term data mining with well-established processes, with limited automation possibilities at the time [151]; and (iii) the human glycome is a highly complex and dynamic cellular component which is being constantly changed over time, possessing a high level of inter-individual variability, so instead of generating a huge amount of information during a single analysis, only the most crucial information (such as aberrant sialylation) should be extracted in an easy and reproducible manner, yielding the relevant clinical information needed for in-time intervention and proper patient management. These analyses would, however, need to be made over an extended period of time to properly determine a “baseline signal” for every individual [152]. These methods, however, could (and definitely should) still become a part of common clinical practice in assessing the risk of a disease being present. For this purpose, high-quality clinical data need to be obtained in large, multinational/multicentric epidemiological clinical studies in the future.

**Table 1 diagnostics-14-00713-t001:** Examples of recently published serum-derived oncomarkers—comparison of miRNAs (miR) as promising and established liquid biopsy biomarkers and different sialic acid-containing (Sia) glycans.

Tumour Location	miRNAs (Combinations)	AUC	Ref.	Sia-Containing Glycans	AUC	Ref.
Bladder	miR-106a-5p, miR-145-5p, miR-132-3p, miR-7-5p, miR-148b-3p	0.922	[153]	Protein-bound Sia	0.825	[154]
Breast	miR15a, miR16	0.884	[155]	Different Sia-glycan isoforms	Up to 0.980	[156]
Colorectum	miR-1246, miR1268b, miR4648	0.821	[157]	H5N4F1, H4N4F1, H5N4F1S_2,6_1	0.830	[158]
Lung	miR-210, miR-1290, miR-150, miR-21-5p	0.930	[159]	Different glycan isoforms + CRP	0.942 *	[160]
Skin (melanoma)	miR-149-3p, miR-150-5p, miR-193a-3p	0.970	[161]	Total serum Neu5Gc	0.925	[162]
Ovaries	miR-92a, miR-200c, miR-320b, miR-320c, miR-335, miR-375, miR-486	0.870	[163]	Total sialylation ratio (α-2,3-Sia) + CA125	0.985 *	[164]
Pancreas	miR-215-5p, miR-122-5p, miR-192-5p, miR-30b-5p, miR-320b	0.811	[165]	Combination of CA4, A3F0L and CFa glycan-isoforms	0.807	[166]
Prostate	miR-4286, miR-27a-3p, miR-29b-3p	0.892 * (+PSA and PV)	[167]	α-2,3-Sia/PSA + PHI	0.985 *	[168]
Stomach	12-miRs panel	0.920	[169]	H5N5F1E2 glycan + other markers	0.892 *	[170]
Testis	miR-371a-3p	0.966	[171]	5 *N*-glycan score	0.870	[172]

H = hexose, N = NAc hexose, F = fucose, AUC = area under the curve parameter from ROC curve, CRP = C-reactive protein, PSA = prostate-specific antigen, PHI = prostate health index, PV = prostate volume; * designates AUC values obtained by combining these markers with those in current use.

Glycomic applications and perspectives in clinical practice have been presented so far in detail; however, the reasons behind the fact that their use in clinical routine is absent could be summarized in a few points:Mass spectrometry (a robust gold standard in structural glycobiology) workflows need to be further transformed to advance the translation to clinics for suitable quantification of glycoconjugates. Mass spectrometry imaging for the observation of spatial distribution of glycans in histological samples is also of eminent importance.From a regulatory point of view, since there is currently a lack of any glycan detection diagnostic kits on the market based on common immunosorbent assays, any new products will be reviewed and judged according to common practices with ELISA. There are some aspects, however, which are unique for Enzyme-Linked Lecin Binding Assay (ELLBA) platforms, such as suppressing non-specificity based on lectin substrate promiscuity and low affinity between the lectins and glycans [173].In the therapeutic area, the first anti-core 1 *O*-glycans monoclonal antibody NEO-201 has been involved and registered in a phase I clinical trial in 2018, showing promising results in the treatment of solid tumours in 2023 [174]. Investments and support from large pharma companies in next rounds of clinical trials could help to accelerate advancement in the area and thus help to bring new molecules into clinical routine much sooner.

## 5. Conclusions

This review focused on the molecular interactions of sialylated glycans in vivo in physiological and pathological processes, their analysis and a comparison with current state-of-the-art methods, especially cutting-edge liquid biopsy approaches, such as miRNAs analysis. We showed that glycans have a number of advantages, including a short pre-analytical phase, short and easy-to-perform analytical protocols in the case of ELLBA formats of analysis in combination with glycan-binding receptors, and high levels of diagnostic accuracy. As yet, glycans are not commonly used in clinical practice due to the lack of analytical standards and due to the need to implement these practices into the high-throughput solutions preferred by clinics. Since any success in cancer therapy and the subsequent quality of a patient’s life is strongly related to the stage of the disease at the time of diagnosis, early-stage diagnostics in combination with the non-invasive nature (to reduce any barriers between patient and clinician) of the glycans analysis and possible organ specificity currently represent an urgent medical need to be met, where glycans can provide a promising tool for decades to come.

## Figures and Tables

**Figure 1 diagnostics-14-00713-f001:**
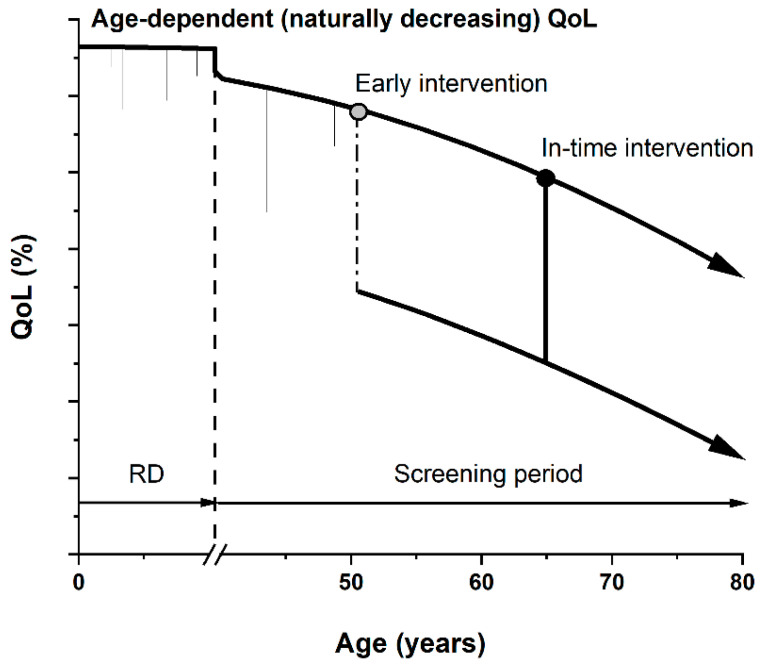
Graphic presentation of quality of life (QoL) parameter as a function of age. QoL naturally decreases with greater age due to various accumulated conditions and poor health. Short-term decreases can also be seen (common injuries or illnesses) during risk development period (RD, ending with dashed line). During the screening period, if an early stage or pre-cancerotic stage is diagnosed, overdiagnosis and overtreatment (starting with a grey circle) might significantly and permanently decrease patients’ QoL (dash-dotted line), i.e., in cases of a radical prostatectomy leading to erectile dysfunction and incontinence. In-time diagnostics and subsequent intervention (starting with a black circle) yield a higher QoL for a longer time period, quantifiable by the area under the curve.

**Figure 2 diagnostics-14-00713-f002:**
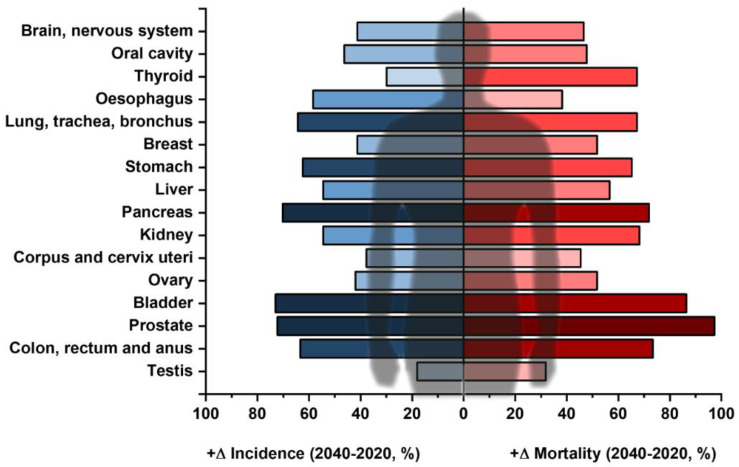
Commonest types of solid tumours (except for lymphomas and skin cancers) and their expected increase in different parameters, i.e., predicted incidence (blue) and predicted mortality (red) according to GLOBOCAN data between 2020 and 2040 (in %). The darker the colour, the more the respective cancer is predicted to increase for the respective parameter. Data taken from ref. [24].

**Figure 4 diagnostics-14-00713-f004:**
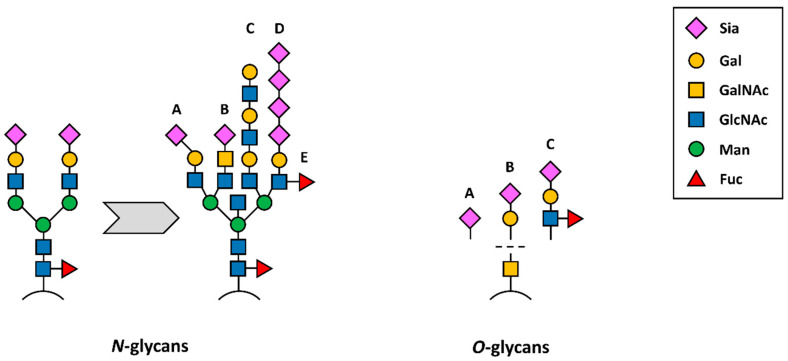
Schematic presentation of *N-*(left) and *O*-glycans (right) detected during tumourigenesis. From a common biantennary complex type *N*-glycan, different structures are derived, such as (A) α2,3-linked Sia, (B) Sia attached to LacdiNAc (sialyllactose diamine), (C) desialylated structure with lactosamine repeats, (D) α2,8-linked polysialic acids and/or (E) antennary fucosylation. *O*-glycans, on the other hand, are truncated during tumourigenesis, leading to the occurrence of Tn antigen (bare GalNAc structure) or (A) sTn, (B) sT antigen and (C) sLewis antigens A or X.

## Data Availability

Not applicable.

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
