# Peer review of "Medical Relevance, State-of-the-Art and Perspectives of “Sweet Metacode” in Liquid Biopsy Approaches"

_diagnostics, 2024, doi:10.3390/diagnostics14070713_

Round 1
Reviewer 1 Report
Comments and Suggestions for Authors
The manuscript is devoted to a new and interesting topic - the role of glycans with terminal sialic acids in the development of oncology. The authors point to the potential of liquid biopsy in the early diagnosis of various types of cancer. At the same time, the authors pay great attention to the possibility of overdiagnosis and, as a result, a deterioration in the quality of life of the patient. Figure 1 shows this clearly.
The manuscript contains a detailed description of the biochemistry of glycans and perspective of aberrant sialylation during tumourigenesis and its significance. The authors describe methods for determining glycans with terminal sialic acids in the blood, point out the difficulties of quantitative determination of these substances and their clinical validations as oncomarkers.
The manuscript is logically organized and understandable, and contains figures useful for understanding of the text. The manuscript fit the journal scope and it is of direct relevance to community of Diagnostics. I think, the manuscript attracts a wide readership and contain interesting ideas.
However, I have some remarks and comments. Therefore, the manuscript may be accepted after minor revision.
My remarks and comments:
1. The manuscript contains many undeciphered abbreviations that make the text difficult to understand, for example:
Line 114 - CEA, AFP
Line 132 – ROS
Line 195 - NK cells
Line 303 – “namely GD2, GD3 and GM2 along with sTn antigen and sialyl- Lewis (sLe) antigens. sLeA (CA19-9) is an FDA-approved oncomarker”
I understand that many of the abbreviations are associated with the specific names of proteoglycans. Maybe in the Supplementary Materials you can provide the structures and names of the most important of them?
2. Line 122 - Warburg effect (observed by Otto Warburg et al. in 1920) - If it is written like this, a reference to the work of 1920 should be given.
3. Fig. 3 and 4 contain many undeciphered abbreviations and therefore not everything is clear.
4. Line 237 - (T+C)/K – What does "K" mean?
5. Line 248 – “Although more complex, if aberrant glycosylation forms a part of an early cs tumour development” What does "cs" mean?
Author Response
The response to the reviewer´s comments is in the attached file.

Reviewer 2 Report
Comments and Suggestions for Authors
Weaknesses:
Organization: The article's structure could be improved for better readability. Some sections, such as the discussion on verification/validation studies and challenges in glycomics, appear somewhat fragmented and could benefit from clearer organization or subheadings.
Technical Jargon: While the article provides a detailed discussion on the topic, it occasionally uses technical language and abbreviations that may be challenging for readers unfamiliar with the field. Providing clearer explanations or definitions for specialized terms could enhance accessibility.
Limited Discussion on Solutions: While the article thoroughly outlines challenges in glycomics and liquid biopsy, it offers limited discussion on potential solutions or future directions to address these challenges. Including suggestions or insights into emerging technologies or methodologies could enrich the discussion.
Overall, while the article provides a comprehensive overview of glycan biomarkers, addressing these weaknesses could further improve its clarity, accessibility, and impact
Comments on the Quality of English Language00
Author Response

(The authors gave the same response as above.)
